# Effect of Kinesiophobia and Social Support on Quality of Life After Total Hip Arthoplasty

**DOI:** 10.3390/healthcare13121366

**Published:** 2025-06-06

**Authors:** Panagio Marmouta, Lemonia Marmouta, Andreas Tsounis, Chara Tzavara, Maria Malliarou, Evangelos Fradelos, Maria Saridi, Aikaterini Toska, Pavlos Sarafis

**Affiliations:** 1Department of Health Care Management, Hellenic Open University, 26335 Patras, Greece; pmarmouta@gmail.com (P.M.); lmarmouta@gmail.com (L.M.); 2Department of Psychology, Aristotle University of Thessaloniki, 54124 Thessaloniki, Greece; atsounis@psy.auth.gr; 3School of Medicine, National and Kapodistrian University of Athens, 11527 Athens, Greece; htzavara@med.uoa.gr; 4Department of Nursing, University of Thessaly, 41500 Larissa, Greece; malliarou@uth.gr (M.M.); efradelos@uth.gr (E.F.); saridi@uth.gr (M.S.); ktoska@uth.gr (A.T.)

**Keywords:** kinesiophobia, social support, quality of life, total hip arthroplasty

## Abstract

**Background/Objectives:** Total hip arthroplasty (THA) improves quality of life in patients with hip osteoarthritis (OA) by alleviating pain and restoring mobility. Kinesiophobia (i.e., fear of performing specific movements to avoid pain and re-injury) negatively affects the quality of life after THA, while social support impacts postoperative quality of life by influencing recovery and well-being. This cross-sectional study investigates the effects of kinesiophobia and social support, as well as their interaction, on the quality of life after THA. **Methods:** A total of 125 patients participated in the study, all of whom had undergone THA for end-stage hip OA. The Tampa Scale for Kinesiophobia (TSK), the Oslo Social Support Scale (OSSS), and the Total Quality of Life scale (T-QoL) were used for the assessment of the study variables. Multiple linear regression was conducted considering the T-QoL subscales (emotional and physical well-being, functional engagement, resilience, and peri-traumatic experience) as dependent variables. **Results**: Kinsesiophobia correlated negatively with physical and emotional well-being, peri-traumatic experience, and resilience. Social support was positively correlated with emotional well-being, functional engagement, and resilience; however, it also buffered the negative relationship between kinesiophobia and peri-traumatic experience. Age was negatively correlated with emotional and physical well-being, functional engagement, and resilience, while the patient being female also correlated negatively with emotional well-being and functional engagement. Finaly, self-perception of mental health problems was negatively correlated with resilience. **Conclusions:** Future studies may further investigate the pathway between kinesiophobia and social support on the one hand and quality of life on the other hand, as well as the interaction between social support and kinesiophobia.

## 1. Introduction

Hip osteoarthritis (OA) is a chronic degenerative joint disease that primarily affects the hip joint, leading to progressive deterioration of cartilage, pain, stiffness, and functional impairment. It is one of the most common causes of disability in older adults and can significantly impact the quality of life. It is more common in individuals over the age of 50 and affects women more frequently than men [1]. Hip OA results from a combination of mechanical stress and biochemical changes that lead to cartilage breakdown, subchondral bone remodeling, osteophyte formation, and synovial inflammation [1]. Risk factors for hip OA include aging, obesity, genetic predisposition, joint injuries, and occupations that involve repetitive stress on the hip joint [2]. Hip OA presents with groin pain, stiffness, a reduced range of motion, crepitus, and joint deformity in advanced cases [2]. X-rays show joint space narrowing, osteophyte formation, and subchondral sclerosis. Non-surgical management includes weight loss, physical therapy, analgesics, non-steroidal anti-inflammatory drugs (NSAIDs), and corticosteroid injections [3]. When conservative treatment fails, surgical options may be considered. Total hip arthroplasty (THA) is the gold standard for end-stage hip OA, involving the replacement of the damaged joint with a prosthetic implant, providing significant pain relief and restoring mobility [4].

THA significantly improves the quality of life by alleviating pain, restoring mobility, and enhancing overall well-being [4]. Patients experience a reduction in chronic discomfort, leading to increased physical activity and an improvement in sleep and mental health [5]. The ability to perform daily tasks without pain fosters independence and social engagement, reducing feelings of isolation [5,6]. However, recovery can be challenging, requiring strong social support, adherence to rehabilitation, and emotional resilience [7]. While most patients report high satisfaction post-THA, outcomes depend on preoperative health, post-surgical care, and lifestyle adjustments [8].

Kinesiophobia is a condition characterized by an excessive, irrational fear of movement due to a perceived risk of injury or pain, leading to avoidant behaviors that can exacerbate physical and psychological distress, ultimately hindering recovery from musculoskeletal injuries and chronic pain conditions [9]. It is commonly observed in patients with conditions such as chronic lower back pain, osteoarthritis, fibromyalgia, and post-surgical recovery. The term originates from the Greek words “kinesis”, meaning movement, and “phobos”, meaning fear, and was first introduced in the field of rehabilitation medicine to describe patients who develop maladaptive movement avoidance behaviors [9,10]. Kinesiophobia after THA is a significant concern that can hinder post-surgical recovery and affect long-term functional outcomes [11]. While the primary goal of THA is to relieve pain and restore mobility, some patients develop kinesiophobia due to concerns about pain, implant failure, or re-injury, leading to avoidance behaviors, muscle deconditioning, and suboptimal rehabilitation outcomes [11,12].

Social support is a fundamental factor influencing quality of life after THA [7,13]. Strong emotional backing, practical assistance, and professional guidance contribute to a smoother recovery, improved functional outcomes, and enhanced psychological well-being [14]. Recovery from THA extends beyond the physical healing of the joint, encompassing emotional well-being, rehabilitation adherence, and overall life satisfaction [15]. A strong social support system, comprising family, friends, caregivers, and healthcare professionals, can positively influence functional recovery, emotional stability, and long-term independence, while a lack of support may hinder progress and lead to poorer outcomes following THA [14]. Importantly, social support could also be considered a buffer in the negative relationship between kinesiophobia and quality of life. Previous research has provided evidence for the buffering role of social support in pain and stressful conditions [14].

Furthermore, several sociodemographic factors may also relate to the quality of life after THA. Research findings regarding gender are contradictory [16]. The findings in a literature review showed that from the fifteen studies that evaluated gender as a predictor, nine did not found statistically significant findings, while in four studies being male was related to better functional outcomes, and in three of the studies the relationship was significant for female patients [16]. Regarding age, the evidence from the literature is more consistent, as most studies show that younger patients recover better and report higher quality of life levels after THA [16]. The evidence regarding other factors such as educational level is limited. The relationship between place of residence and quality of life after THA has also not been widely studied, although it may determine factors such as access to health services and to further resources.

The aim of this study is to investigate the relationship between kinesiophobia, social support, and several sociodemographic features one the one hand and quality of life after THA on the other, as well as the moderating role of social support in the relationship between kinesiophobia and quality of life. Kinesiophobia and social support may trigger two different processes regarding quality of life after a surgery. Kinesiophobia, on the one hand, is negatively correlated with several quality of life aspects—it delays post-surgery recovery through activity avoidance that, beyond physical performance, may also impair emotional well-being as well as the sense of resilience [17]. Social support, on the other hand, may help patients enhance their functionality in everyday life, while also being positively correlated to better emotional health perceptions during the recovery process [13,18,19]. By testing these parallel processes in kinesiophobia and social support in relation with quality of life after THA, the current study may shed light on two parallel processes, a negative process (beginning with kinesiophobia) and a positive process (beginning with social support), by providing evidence of these relationships in the same population. This is an important advancement, as it may broaden the finding of the previous studies that examine these relationships separately. Moreover, by testing the buffering role of social support in the negative relationship between kinesiophobia and quality-of-life, we may shed more light on their potential interaction in the recovery process after THA. Finally, by taking account several sociodemographic features, we may validate previous findings (e.g., from studies on gender or age) and expand the knowledge regarding the aspects (e.g., place of residence) that are related to quality of life after THA and that have not been studied systematically.

## 2. Materials and Methods

### 2.1. Settings and Procedure

This study comprises institution-based, cross-sectional research that was conducted at KAT Hospital, Athens, Greece. The study population consisted of a convenience sample. Patients from that hospital who had THA were invited to participate in the study. Data were collected between June 2021 and February 2022; at the time of the survey each participant has completed a five-month of a post-operative period, so that the surgery could be considered eligible. Inclusion criteria were as follows: adult patients subjected to elective THA due to hip OA, with sufficient knowledge of Greek language (i.e., in a possible case of immigrant participants). Exclusion criteria were as follows: the absence of a diagnosis of mental and/or cognitive impairment problems. The study population consisted of 125 participants (response rate: 83.33%).

### 2.2. Measures

The data for this study were collected through a questionnaire comprised of four sections. Section 1 included patient demographic data and baseline characteristics (i.e., gender, age, education level, place of residence, working status, marital status, occupation, and individual perception of presence of mental and/or physical health problem). Section 2 included the Total Quality of Life scale (T-QoL) [20] which assessed post-injury quality of life. The T-QoL scale is a tool designed to assess the overall quality of life by measuring multiple dimensions of well-being (i.e., physical, psychological, social, and environmental factors that influence an individual’s perception of life satisfaction). Namely, T-QoL subscales include emotional well-being (e.g., “I feel safe on a day-to-day basis”), functional engagement (e.g., “I need help bathing/showering”), recovery/resilience (e.g., “My recovery was shorter than I expected”), peri-traumatic experience (e.g., “Overall, the care I got in the hospital was good”), and physical well-being (e.g., “I have pain on a daily basis”) [20]. The choice of T-QoL compared to generic instruments was made because T-QoL could be perceived as more sensitive in the assessment of quality of life after a chronic disease such as OA that presents similarities to injury conditions [21], since its items are more specifically anchored to patients’ perceptions of injury, which in turn may facilitate the interpretation of how an injury or a surgery may affect the quality of life [22]. Section 3 assessed kinesiophobia with the Tampa Scale for Kinesiophobia (TSK) [23], which is one of the most widely used tools for its evaluation. The scale consists of seventeen statements assessing an individual’s fear of movement and re-injury, with higher scores indicating greater levels of kinesiophobia. An example item of the scale is “I am afraid I might injure myself if I exercise” [23]. Section 4 assessed social support with the Oslo Social Support scale (OSSS-3) [24]. The OSSS-3 is a brief and widely used tool which consists of three items that evaluate the availability and quality of social relationships, including the number of close contacts (e.g., “How many people are so close to you that you can count on them if you have great personal problems?”), the level of concern shown by others, and the perceived ease of receiving practical help [25]. Its psychometric properties have been tested both in the general population [25], as well as in more specific populations like older adults [25], and thus it is considered an adequate tool for social support assessment among patients with OA.

### 2.3. Ethical Issues

Our study protocol received approval from the Institutional Review Board of General Hospital of Athens “KAT” (approval number 648; approved 4 June 2021) and adhered to the principles of the Declaration of Helsinki. The completion of the survey was voluntary and anonymous. All participants provided informed consent prior to enrollment and had the right to withdraw at any time during the data collection.

### 2.4. Statistical Analysis

Quantitative variables were expressed as mean values and standard deviations (SDs), while qualitative variables were expressed as absolute and relative frequencies. Multiple linear regression analysis was used with dependent the T-QoL subscales. The regression equation included terms for participants’ characteristics, levels of kinesiophobia and levels of social support. Adjusted regression coefficients (β) with 95% confidence intervals (95% CI) were computed from the results of the linear regression analyses. Diagnostics for regression models were performed to check if the conditions for regression had been met with the residuals of each model being normally distributed and their variance being constant. Multicollinearity was assessed using variance inflation factor (VIF). For the investigation of the moderating role of social support scale in the association between the kinesiophobia scale and QoL, the SPSS PROCESS macro was used following the Hayes guidelines [26]. In case of a significant interaction, two-point estimates (−1SD, +1SD) of the social support scale were calculated. All reported *p* values were two-tailed. Statistical significance was set at *p* < 0.05 and analyses were conducted using SPSS statistical software (version 26.0).

## 3. Results

Sample characteristics are presented in Table 1. Most participants were women (60%), with 57.6% being under 70 years old and 61.6% being middle/high school graduates. Most participants were married (67.2%) and 60.8% were living in the capital. Moreover, 56.8% were pensioners, 8.0% reported a mental health problem and 76.8% a physical health problem.

The mean score for kinesiophobia was 10.4 (SD = 2.5) and for social support it was 44.8 (SD = 9.5). Table 2 presents means, standard deviations, minimum and maximum scores, as well as Cronbach’ s alpha for the T-QoL sub-scales.

After multiple linear regression analysis, it was found that participants who were 70–90 years old and female had significantly lower emotional well-being and functional engagement scores compared to participants who were younger than 70 years old and male, respectively (Table 3). Also, kinesiophobia was significantly and negatively correlated with emotional well-being, while social support was significantly and positively correlated with emotional well-being and functional engagement.

Recovery/resilience score was significantly and negatively correlated with age, being lower in participants who were 70–90 years old and who reported a mental health problem and kinesiophobia, while it was positively correlated with social support. Kinesiophobia was also negatively related with peri-traumatic experience (Table 4).

Physical well-being score was significantly associated with the participant’s age and kinesiophobia (Table 5). Namely, it was significantly lower in participants who were 70–90 years old, while kinesiophobia was also negatively correlated to physical well-being.

Finally, we examined the moderating role of social support in the relationship between kinesiophobia and QoL sub-scales. We found a significant moderating role only between kinesiophobia and peri-traumatic experience (*p*_interaction_ = 0.003). Namely, as social support increased, the negative effect of kinesiophobia on peri-traumatic experience decreased (Figure 1). In the rest of the QoL scales, we found no moderating effect of social support.

## 4. Discussion

The results of the present study showed that post-injury quality of life after THA is associated with the patients’ gender, age, social support, mental health, and kinesiophobia. Kinsesiophobia correlated negatively with physical and emotional well-being, peri-traumatic experience, and resilience, while social support was positively correlated with emotional well-being, functional engagement, and resilience. Social support also buffered the negative relationship between kinesiophobia and peri-traumatic experience. Being female also correlated negatively with emotional well-being and functional engagement, while age was negatively correlated with emotional and physical well-being, functional engagement, and resilience. Finaly, self-perception of a mental health problem was negatively correlated with resilience.

Kinesiophobia related negatively and significantly with almost all sub-scales of T-QoL. The negative effects of kinesiophobia on both emotional and physical aspects of well-being, as well as on the negative perceptions of peri-traumatic experience have been documented in previous studies [10,11,12]. Resilience, on the other hand, although it has been approached either as a trait (i.e., personal characteristic that helps individuals cope with adversity), an outcome (i.e., behavioral function that enhances recovery), and/or a process (i.e., dynamic process in which individuals actively adapt to and recover from adversities), in all cases is perceived as a core and dynamic aspect of the ability for recovery [27]. Thus, the potential negative effect of kinesiophobia on resilience may deteriorate the long-term recovery process, since resilience, despite being considered a trait or a skill that is open to change, may keep individuals highly motivated when they face chronic and stressful conditions [27].

According to our findings, social support was correlated positively and significantly with emotional well-being, functional engagement, and resilience. Patients with strong social support systems tend to recover better, since they may have better access to resources, while a social context of supportiveness may help them to adopt positive coping measures, enhance their ability to deal with stress, and increase their self-efficacy during the rehabilitation [28]. Assistance with transportation, household chores, meal preparation, and encouragement to adhere to rehabilitation exercises can make a significant difference [29]. Moreover, patients with stronger social support systems, who are more socially engaged, usually desire to return to their social activities, which is something that can motivate them to confront the obstacles to rehabilitation more actively [28]. Considering the above, social support, beyond its direct potential positive effects on several aspects of quality-of-life, can also function as a protective factor in stressful and adverse situations that are related to peri-traumatic experiences [19], in line with our findings.

Our study also revealed a statistically significant relationship with aspects of quality of life and both age and gender. Age is crucial for post-THA recovery. Study results revealed that age correlated negatively with emotional and physical well-being, as well as, functional engagement, and resilience. These findings are in line with empirical evidence and suggest that younger patients face both short-term [30] and long-term [31] positive post-operative functional outcomes. Younger patients generally have better muscle strength, joint flexibility, and overall physical resilience, allowing for a faster return to daily activities. They are more likely to engage in physical therapy, adhere to rehabilitation protocols, and regain functional independence quickly [32]. Moreover, our findings showed a negative relationship between age and both emotional well-being and resilience. Older patients face more adverse effects during their recovery (e.g., presence of comorbidities such as cardiovascular disease, diabetes, and arthritis), while they also have less opportunities to perform physical and social activity, which in turn may negatively affect their sense of independence, resilience and physical and emotional aspects of well-being [33]. Regarding the role of gender, findings of the relevant literature are contradictory [16]. However, most studies support that being female is related to worse post-surgery outcomes after THA, in line with our findings [16].

Finaly, the results showed that recovery/resilience after THA was significantly lower for those who reported a mental health problem. Mental health problems can significantly affect postoperative quality of life after THA by influencing pain perception and rehabilitation adherence, which may lead to prolonged recovery times, reduced functional improvement, and lower satisfaction with surgery [34]. The above may negatively affect the sense of resilience. However, for the interpretation of this finding we have to take into account that in the current study mental health problems were not based on a diagnosis, which may affect the robustness of the finding. Similarly, in all the relationships which are described above we cannot ignore the potential impact of reverse causality (e.g., patients with a lower quality of life may perceive less support, greater kinesiophobia or mental and physical health problems).

The above findings may lead to practical implications that may facilitate post-operative recovery. The most common approach for the management of kinesiophobia is based on physical exercise, which may include progressive muscle relaxation, pilates, and body awareness exercises [35]. However, given that kinesiophobia is more than a simple fear of movement, but rather a multifactorial mindset which stems from the belief of fragility [36], which in our study is negatively correlated with resilience, interventions may include psychological support, which may incorporate mindfulness interventions, pain and stress management education, and relaxation training [35]. Importantly, interventions may also include the cooperation of more than one specialty, such as physiotherapists, whose role is crucial in the post-operative period, medical staff, providing the adequate patient-centered medication for pain relief (i.e., based on his/her personal characteristics such as age and comorbidity), as well as nursing staff who provide peri-operative care in the initial stage after surgery [35]. Beyond the management of kinesiophobia, quality of life after THA may be improved through the enhancement of social support, which in our study was positively correlated with several quality of life aspects after the surgery, while it also functioned as a potential moderator of the negative relationship between kinesiophobia and peri-traumatic experience. Relevant interventions may incorporate emotional and informational support from health professionals (i.e., provide relevant information, telephone coaching), and group-based interventions (i.e., group discussion for promoting empathy and sharing coping techniques between members), that may reinforce adaptive behaviors through positive peer-interactions, in contrast to individual-level interventions, that, according to research findings, are less effective [18].

The current study has several limitations. First, the lack of data regarding the preoperative health condition, as well as quality of life and social support, may weaken the potential conclusions of the study and limit causal inference. Second, there are no data about the preoperative clinical and radiological severity of hip OA, as well as surgical details, such as surgical approaches and rehabilitation regimes, which have a major role impact in THA clinical outcome. Third, self-reported data may lead to common-method variance problems. Forth, T-QoL, TSK and OSS-3 has not been validated in the Greek population. Finaly, the study’s cross-sectional design and the fact that the participants came from a single hospital, which may lead to potential selection bias, diminishes the potential for the generalization of the findings.

## 5. Conclusions

Post-injury quality of life after THA is associated with patients’ social support, kinesiophobia, as well as several sociodemographic features. Kinsesiophobia related negatively with physical and emotional well-being, peri-traumatic experience and resilience, while social support related positively with emotional well-being, functional engagement, and resilience. Importantly, the negative relationship between kinesiophobia and peri-traumatic experience was lower for patients with stronger social support. Meanwhile, age was negatively correlated with several quality of life aspects, while being a woman also related negatively with emotional well-being and functional engagement. Future studies, by implementing a longitudinal design, may further investigate the effects of kinesiophobia on quality of life by also testing aspects, such as social support, that may buffer this relationship, as well as the pathway from social support to emotional wellbeing by fostering the specific mechanisms that boost the recovery procedure after THA.

## Figures and Tables

**Figure 1 healthcare-13-01366-f001:**
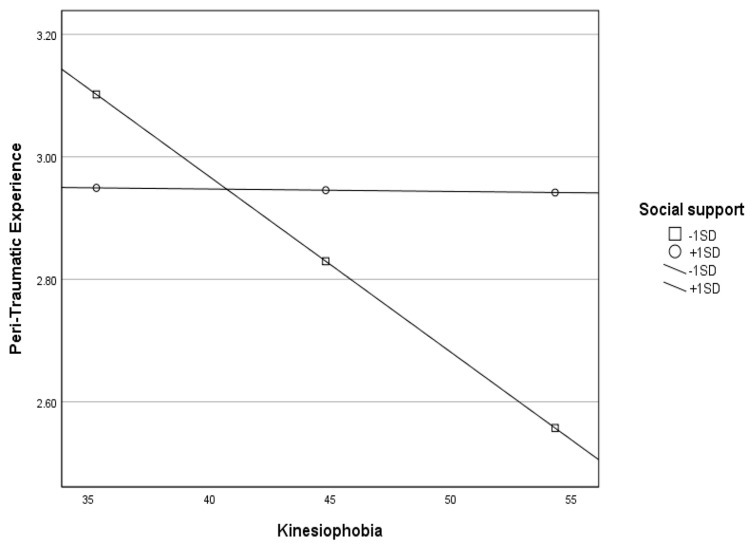
Moderating effect of the social support scale on the relationship between kinesiophobia and peri-traumatic experience scale.

**Table 1 healthcare-13-01366-t001:** Sample characteristics (N = 125).

	n	%
Gender		
Men	50	40.0
Women	75	60.0
Age (years)		
<70	72	57.6
70–90	53	42.4
Educational level		
Middle/High school	77	61.6
University	48	38.4
Married	84	67.2
Living alone	35	28.0
Living in the capital	76	60.8
Working status		
Part/Full time employee	19	15.2
Household/Unemployed	35	28.0
Pensioner	71	56.8
Mental health problem	10	8.0
Physical health problem	96	76.8
Kinesiophobia		
Low	31	24.8
High	94	75.2
Social support levels (OSLO)		
Low	22	17.6
Moderate	58	46.4
High	45	36.0

**Table 2 healthcare-13-01366-t002:** Descriptive measures for T-QoL subscales.

	Minimum	Maximum	Mean (SD)	a
Emotional Well-Being	1.56	4.00	2.77 (0.63)	0.90
Functional Engagement	1.63	4.00	3.05 (0.70)	0.89
Recovery/Resilience	1.00	4.00	2.66 (0.80)	0.89
Peri-Traumatic Experience	1.20	4.00	2.85 (0.56)	0.73
Physical Well-Being	1.00	3.75	2.24 (0.72)	0.87

Note: Greater values indicate better QoL.

**Table 3 healthcare-13-01366-t003:** Multiple linear regression results with emotional well-being and functional engagement as dependent variables.

		Emotional Well-Being			Functional Engagement		
		Β +	95% CI ++	*p*	Β +	95% CI ++	*p*
Gender	Men (reference)						
	Women	−0.25	−0.46–−0.04	0.02	−0.27	−0.53–−0.01	0.04
Age (years)	<70 (reference)						
	70–90	−0.27	−0.48–−0.05	0.01	−0.58	−0.85–−0.32	0.00 **
Educational level	Middle/High school (reference)						
	University	−0.02	−0.22–0.19	0.87	−0.15	−0.40–0.10	0.24
Married	No (reference)						
	Yes	−0.11	−0.36–0.13	0.35	−0.13	−0.43–0.18	0.41
Living alone	No (reference)						
	Yes	0.01	−0.25–0.28	0.93	0.15	−0.18–0.48	0.38
Living in the capital	No (reference)						
	Yes	0.09	−0.10–0.29	0.35	−0.03	−0.27–0.22	0.84
Working status	Part/Full time employee (reference)						
	Household/Unemployed	0.15	−0.09–0.39	0.21	0.12	−0.18–0.41	0.43
	Pensioner	−0.11	−0.40–0.17	0.43	−0.20	−0.55–0.16	0.27
Mental health problem	No (reference)						
	Yes	−0.13	−0.50–0.24	0.49	−0.38	−0.84–0.07	0.10
Physical health problem	No (reference)						
	Yes	−0.07	−0.29–0.15	0.53	−0.20	−0.47–0.08	0.16
Kinesiophobia		−0.02	−0.03–−0.01	0.00 **	0.00	−0.01–0.01	0.94
Social support		0.08	0.03–0.13	0.00 *	0.06	0.00–0.12	0.04
Adjusted R^2^		0.38			0.24		

* *p* value < 0.01; ** *p* value < 0.001; + regression coefficient; ++ 95% Confidence Interval.

**Table 4 healthcare-13-01366-t004:** Multiple linear regression results with recovery/resilience and peri-traumatic experience as dependent variables.

		Recovery/Resilience			Peri-Traumatic Experience		
		Β +	95% CI ++	*p*	Β +	95% CI ++	*p*
Gender	Men (reference)						
	Women	0.06	−0.20–0.33	0.63	−0.13	−0.35–0.10	0.26
Age (years)	<70 (reference)						
	70–90	−0.41	−0.67–−0.14	0.00 *	−0.03	−0.25–0.20	0.82
Educational level	Middle/High school (reference)						
	University	−0.05	−0.30–0.20	0.70	−0.01	−0.23–0.20	0.90
Married	No (reference)						
	Yes	−0.08	−0.39–0.22	0.58	0.13	−0.13–0.38	0.32
Living alone	No (reference)						
	Yes	0.05	−0.28–0.37	0.77	0.00	−0.27–0.28	0.98
Living in the capital	No (reference)						
	Yes	0.00	−0.24–0.25	0.99	−0.07	−0.28–0.14	0.49
Working status	Part/Full time employee (reference)						
	Household/Unemployed	−0.01	−0.30–0.28	0.95	−0.13	−0.38–0.11	0.28
	Pensioner	0.11	−0.24–0.47	0.53	0.01	−0.29–0.31	0.96
Mental health problem	No (reference)						
	Yes	−0.58	−1.03–−0.12	0.01	−0.18	−0.57–0.20	0.35
Physical health problem	No (reference)						
	Yes	0.07	−0.20–0.35	0.59	0.06	−0.17–0.29	0.61
Kinesiophobia		−0.03	−0.04–−0.02	0.00 **	−0.02	−0.03–0.00	0.01
Social support levels		0.10	0.04–0.16	0.00 *	0.04	−0.01–0.09	0.10
Adjusted R^2^		0.42			0.15		

* *p* value < 0.01; ** *p* value < 0.001; + regression coefficient; ++ 95% Confidence Interval.

**Table 5 healthcare-13-01366-t005:** Multiple linear regression results with physical well-being as a dependent variable.

		Physical Well-Being		
		Β +	95% CI ++	*p*
Gender	Men (reference)			
	Women	0.02	−0.24–0.29	0.87
Age (years)	<70 (reference)			
	70–90	−0.55	−0.82–−0.29	0.00 **
Educational level	Middle/High school (reference)			
	University	−0.17	−0.42–0.08	0.19
Married	No (reference)			
	Yes	0.04	−0.26–0.35	0.79
Living alone	No (reference)			
	Yes	0.20	−0.13–0.53	0.23
Living in the capital	No (reference)			
	Yes	0.00	−0.24–0.25	0.98
Working status	Part/Full time employee (reference)			
	Household/Unemployed	0.07	−0.23–0.36	0.64
	Pensioner	−0.15	−0.51–0.20	0.40
Suffer from a mental health problem	No (reference)			
	Yes	−0.25	−0.71–0.21	0.28
Suffer from a physical health problem	No (reference)			
	Yes	−0.19	−0.47–0.08	0.17
Kinesiophobia		−0.02	−0.03–0.00	0.02
Social support levels		0.05	−0.01–0.11	0.08
Adjusted R^2^		0.27		

** *p* value < 0.001; + regression coefficient; ++ 95% Confidence Interval.

## Data Availability

Data can be obtained from the corresponding author.

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
