# Peer review of "Effect of Kinesiophobia and Social Support on Quality of Life After Total Hip Arthoplasty"

_healthcare, 2025, doi:10.3390/healthcare13121366_

Round 1
Reviewer 1 Report
Comments and Suggestions for Authors
Dear authors,
Thank you for the opportunity to review your manuscript titled Effect of Kinesiophobia and Social Support on Quality of Life after Total Hip Arthroplasty. The study addresses an important area in orthopedic and psychosocial rehabilitation, exploring the associations between psychological factors and patient-reported outcomes following total hip arthroplasty (THA). While the topic is timely and relevant, the manuscript would benefit from a clearer conceptual framework, more rigorous methodology reporting, and a more focused interpretation of findings. Below are detailed comments and suggestions for improvement.
abstract
The abstract appropriately introduces the topic, objective, methods, and main findings. However, it could be improved by avoiding vague expressions such as "had lower" or "reported greater" without specifying whether these differences were statistically significant and clinically meaningful. Indicate which T-QoL subdomains were significantly associated with kinesiophobia and social support, and quantify the magnitude of these effects. Clarify that the study is cross-sectional and does not establish causal relationships.
introduction
The introduction provides adequate background on hip osteoarthritis, THA, and their impact on quality of life. The explanations of kinesiophobia and social support are conceptually appropriate but would benefit from a more focused integration into the rationale. The current version reads like three parallel narratives (OA, kinesiophobia, social support) rather than a cohesive justification for studying their combined effect on post-THA outcomes. The literature cited is current and relevant, but the research gap is not sharply defined. Explain why previous studies have not sufficiently explored the role of both kinesiophobia and social support together in this population and how this study advances existing knowledge.
methods
The methodology section is generally well structured but lacks critical detail needed to assess the study’s validity.
- The study is described as cross-sectional, which is appropriate given the data collection method. However, the timing of the assessments in relation to the surgery (e.g., how long after THA) is not specified. This is crucial, as psychosocial responses and quality of life evolve over time.
- The inclusion and exclusion criteria are insufficiently defined. For example, were patients with cognitive impairment excluded? What was the minimum time since surgery to be eligible?
- The T-QoL scale is introduced with a general description but no citation to a validation paper. It is unclear whether this instrument has been validated in the Greek population or in orthopedic populations.
- The same issue applies to the OSSS-3. Clarify whether its psychometric properties are appropriate for this sample.
- Kinesiophobia is assessed using the TSK, which is appropriate. However, the cutoff used to classify low vs high levels is not stated or justified.
- The decision to categorize continuous predictors like social support or kinesiophobia into dichotomous groups (e.g., low/high) reduces statistical power and may obscure gradients of effect. Consider analyzing these as continuous variables or providing justification for the categorization.
statistical analysis
The analytical approach is reasonable for the study’s objectives, but more information is needed to assess robustness.
- The choice of multiple linear regression models is appropriate, but it is unclear whether multicollinearity was checked among predictors.
- Report R-squared values and model fit statistics for each regression to understand how much variance is explained.
- No correction for multiple testing is mentioned despite running five separate regression models for each T-QoL subdomain.
- The regression results include standard errors and p-values, which is helpful. However, no confidence intervals are presented. Include 95% CIs for each coefficient.
- Several variables (e.g., education level, working status) appear in the regression but are not justified in the conceptual framework. Specify whether these were included based on theory or empirical relevance.
results
The descriptive results are clearly presented. However, the narrative presentation of regression results could be improved by focusing on clinically relevant findings and reducing repetition.
- Age appears as the strongest predictor across multiple outcomes. Discuss whether this reflects functional recovery, psychological resilience, or both.
- The association between high social support and emotional well-being is meaningful. Provide more context on what constitutes "high" support in OSSS-3 terms.
- Kinesiophobia was only significantly associated with recovery/resilience, which contrasts with the emphasis placed on it in the title and abstract. Clarify this discrepancy and consider adjusting the framing of the study’s contribution accordingly.
- The finding that patients living in the capital reported better emotional well-being is interesting but underexplored. Include possible socioeconomic, healthcare access, or environmental explanations.
discussion
The discussion section interprets the findings broadly but would benefit from greater critical reflection and conciseness.
- The authors correctly note that younger age and strong social support are associated with better QoL, consistent with the literature. However, the discussion is overly descriptive and lacks depth in interpreting the psychosocial mechanisms involved.
- There is an overemphasis on location-based differences (capital vs non-capital) without adequate theoretical or empirical grounding.
- The limitations are acknowledged but should be expanded. For instance, the use of self-reported data, potential selection bias due to recruitment from a single hospital, and the absence of preoperative baseline data all limit causal inference.
- The potential impact of reverse causality (e.g., patients with lower QoL perceiving less support or greater kinesiophobia) should be discussed more clearly.
- The recommendation for clinical practice should be more measured. The findings support further study of these factors in post-THA recovery but do not justify direct changes in clinical pathways yet.
conclusion
The conclusion summarizes the main findings appropriately but overstates the implications. Use more tentative language and avoid implying causality. The statement that THA leads to better QoL regardless of age contradicts earlier findings in the manuscript. Emphasize that psychological and social factors moderate recovery outcomes.
Author Response
Response to Reviewer 1 Comments
Summary
Thank you very much for taking the time to review this manuscript. Please find the detailed responses below and the corresponding revisions highlighted in the re-submitted files.
Point-by-point response to Comments and Suggestions for Authors
Comments 1: Dear authors, Thank you for the opportunity to review your manuscript titled Effect of Kinesiophobia and Social Support on Quality of Life after Total Hip Arthroplasty. The study addresses an important area in orthopedic and psychosocial rehabilitation, exploring the associations between psychological factors and patient-reported outcomes following total hip arthroplasty (THA). While the topic is timely and relevant, the manuscript would benefit from a clearer conceptual framework, more rigorous methodology reporting, and a more focused interpretation of findings. Below are detailed comments and suggestions for improvement.
Response 1: We are grateful for your suggestions that have allowed us to make many improvements regarding the theoretical framework, the methods and the interpretation of the findings of the study. Our responses to your comments are shown point by point below. All changes in the manuscript are highlighted in red.
Comment 2: Abstract - The abstract appropriately introduces the topic, objective, methods, and main findings. However, it could be improved by avoiding vague expressions such as "had lower" or "reported greater" without specifying whether these differences were statistically significant and clinically meaningful. Indicate which T-QoL subdomains were significantly associated with kinesiophobia and social support, and quantify the magnitude of these effects. Clarify that the study is cross-sectional and does not establish causal relationships.
Response 2: Thank you for your comment. We rewrote the results in the abstract (lines 27-34). We avoided the expressions that you mention and we focused only on the statistically significant data regarding the relationships between the independent variables and T-QoL subscales. We added also in the abstract the information regarding the cross-sectional nature of the study (Line 20). Further, we took into account your comment regarding the causal relationships in the total manuscript, in order to avoid expressions that imply causality.
Comment 3: Introduction - The introduction provides adequate background on hip osteoarthritis, THA, and their impact on quality of life. The explanations of kinesiophobia and social support are conceptually appropriate but would benefit from a more focused integration into the rationale. The current version reads like three parallel narratives (OA, kinesiophobia, social support) rather than a cohesive justification for studying their combined effect on post-THA outcomes. The literature cited is current and relevant, but the research gap is not sharply defined. Explain why previous studies have not sufficiently explored the role of both kinesiophobia and social support together in this population and how this study advances existing knowledge.
Response 3: Thank you for your comments. We rewrote and expanded the last paragraph of the introduction, where we present the aims of the study (Lines 102-122). There, we tried to explain better our contribution and to provide more specific arguments for the added value of our study, by testing two different “processes” (i.e., a negative which begins from kinesiophobia, and a negative which begins from social support) in the same population. Moreover, we also tested the interaction between social support and kinesiophobia, to test the buffering effects of social support in the negative relationship between kinesiophobia and aspects of quality of life.
Comment 4: methods - The methodology section is generally well structured but lacks critical detail needed to assess the study’s validity. The study is described as cross-sectional, which is appropriate given the data collection method. However, the timing of the assessments in relation to the surgery (e.g., how long after THA) is not specified. This is crucial, as psychosocial responses and quality of life evolve over time.
Response 4: Thank you very much for pointing this. It the Settings and procedure sub-section we clarified the time of the assessments by adding the information: “Data were collected between June of 2021 and February of 2022, while at the time of the survey each participant has completed a five-month of a post-operative period, so as the surgery could be perceived eligible” (Lines 130-132).
Comment 5: The inclusion and exclusion criteria are insufficiently defined. For example, were patients with cognitive impairment excluded? What was the minimum time since surgery to be eligible?
Response 5: Thank you for your suggestion. Inclusion and exclusion criteria were more sufficiently pointed in the text: “Inclusion criteria were as follows: adult patients subjected to elective THA due to hip OA, with sufficient knowledge of Greek language (i.e., in a possible case of immigrant participants). Exclusion criteria were: the absence of a diagnosis of mental and/or cognitive impairment problems”. The minimum time after surgery was pointed above (see the previous comment) (Lines 132-136).
Comment 6: The T-QoL scale is introduced with a general description but no citation to a validation paper. It is unclear whether this instrument has been validated in the Greek population or in orthopedic populations.
Response 6: Indeed, the description of T-QoL is too general. Thus, we added one example item for each sub-scale (Lines 147-156). The scale has not been validated in Greece, and, thus, we mentioned in the limitations (Lines 330-331). Moreover, in the measures subsection we add an argument for the adequacy of its use in comparison with other generic scales that measure quality of life (Lines 152-156). Finally, in the results sub-section (Table 2) we provided the Cronbach a values of T-QoL sub-scales.
Comment 7: The same issue applies to the OSSS-3. Clarify whether its psychometric properties are appropriate for this sample.
Response 7: OSS-3 is a brief instrument that may assess the level of social support in different populations. its psychometric properties have been tested in general population (Kocalevent et al. 2018), while it has been also used among specific populations such as older adults (Bøen et al., 2012), as in our study (see Lines 166-169). We added this information in the text to justify its use in our research. However, since it has not been validated in Greek population, we add this information in the limitations (Lines 330-331).
Comment 8: Kinesiophobia is assessed using the TSK, which is appropriate. However, the cutoff used to classify low vs high levels is not stated or justified.
The decision to categorize continuous predictors like social support or kinesiophobia into dichotomous groups (e.g., low/high) reduces statistical power and may obscure gradients of effect. Consider analyzing these as continuous variables or providing justification for the categorization.
Response 8: Thank you for your comment. We run again the analyses. Kinesiophobia and social support were analyzed as continuous variables and the new results are presented (see Tables 3 to 5).
Comment 9: statistical analysis - The analytical approach is reasonable for the study’s objectives, but more information is needed to assess robustness. The choice of multiple linear regression models is appropriate, but it is unclear whether multicollinearity was checked among predictors.
Response 9: Thank you for the comment. Multicollinearity was assessed using variance inflation factor (VIF) and the information was added in the analysis section (Line 189).
Comment 10: Report R-squared values and model fit statistics for each regression to understand how much variance is explained. No correction for multiple testing is mentioned despite running five separate regression models for each T-QoL subdomain.
Response 10: R-squared values were added. No correction for multiple testing was performed since the statistical tests were independent.
Comment 11: The regression results include standard errors and p-values, which is helpful. However, no confidence intervals are presented. Include 95% CIs for each coefficient.
Response 11: 95% Cis were added in the regression results.
Comment 12: Several variables (e.g., education level, working status) appear in the regression but are not justified in the conceptual framework. Specify whether these were included based on theory or empirical relevance.
Response 12: Thank you for the comment. Although the focus of the study is on kinesiophobia and social support, it is important to present the relevance of sociodemographic that appear in the regression. So, in the introduction sub-section we added a paragraph (before the last one paragraph), in which we provide some basic arguments regarding the inclusion of sociodemographic as independent variables (Lines 91-101).
Comment 13: results - The descriptive results are clearly presented. However, the narrative presentation of regression results could be improved by focusing on clinically relevant findings and reducing repetition.
Response 13: Thank you for your suggestions. By following your recommendation, we make several changes. Now, the results focus more on clinically relevant findings.
Comment 14: Age appears as the strongest predictor across multiple outcomes. Discuss whether this reflects functional recovery, psychological resilience, or both.
Response 14: Thank you for the suggestion. We rewrote the paragraph of the discussion regarding age (Lines 276-288). By following your suggestion, we tried to more clearly present the different aspects of recovery that are related to age and to provide some main arguments for the mechanisms that contribute to recovery.
Comment 15: The association between high social support and emotional well-being is meaningful. Provide more context on what constitutes "high" support in OSSS-3 terms.
Response 15: In the new analyses, social support was analyzed as continuous variable. Thus, the dichotomous groups (e.g., low/high) is no longer presented.
Comment 16: Kinesiophobia was only significantly associated with recovery/resilience, which contrasts with the emphasis placed on it in the title and abstract. Clarify this discrepancy and consider adjusting the framing of the study’s contribution accordingly.
Response 16: In the new analyses, with kinesiophobia and social support as continuous and not dichotomous variables, our findings were partly changed. Kinissiophobia was negatively and significantly related with almost all quality-of-life aspects (i.e., physical and emotional well-being, peri-traumatic experience and resilience). Based to these findings, we made several changes in the discussion.
Comment 17: The finding that patients living in the capital reported better emotional well-being is interesting but underexplored. Include possible socioeconomic, healthcare access, or environmental explanations.
Response 17: In the new analyses living in the capital was no significant in our model. Thus, this paragraph was totally erased, since there was no need to provide justification and interpretation in this finding.
Comment 18: discussion - The discussion section interprets the findings broadly but would benefit from greater critical reflection and conciseness. The authors correctly note that younger age and strong social support are associated with better QoL, consistent with the literature. However, the discussion is overly descriptive and lacks depth in interpreting the psychosocial mechanisms involved.
Response 18: Thank you for the comment. We reformed the discussion by following the order bellow: First, we summarized the findings (1st paragraph), then we discussed our findings, beginning from the relationship between quality-of-life aspects and kinesiophobia, social support sociodemographic, and mental health (2nd to 5th paragraph), next the practical implications, based on our findings (6th paragraph) and finally the limitations (7th paragraph). In the discussion we tried to highlight the mechanisms that explains the significant relationships and provide more specific arguments regarding the interpretation of each finding
Comment 19: There is an overemphasis on location-based differences (capital vs non-capital) without adequate theoretical or empirical grounding.
Response 19: See the comment above. This paragraph was totally erased.
Comment 20: The limitations are acknowledged but should be expanded. For instance, the use of self-reported data, potential selection bias due to recruitment from a single hospital, and the absence of preoperative baseline data all limit causal inference.
Response 20: Thank you for your suggestions. We expanded the limitations by adding the proposed information (Lines 330-334).
Comment 21: The potential impact of reverse causality (e.g., patients with lower QoL perceiving less support or greater kinesiophobia) should be discussed more clearly.
Response 21: Thank you for the comment. The study design set the quality-of-life as an outcome. Analysis and discussion were based on this. If we discuss with details the potential impact of reverse causality we will partly go beyond the main hypotheses of the study. However, we mentioned in the text that the consideration of reverse causality is important.
Comment 22: The recommendation for clinical practice should be more measured. The findings support further study of these factors in post-THA recovery but do not justify direct changes in clinical pathways yet.
Response 22: Thank you for your comment. At the end of the discussion (before limitations), we added a new paragraph with practical implications (Lines 303-324). There, we tried to provide some more details regarding potential interventions which may enhance quality of life after THA, through kinesiophobia management and social support enhancement. In the text we tried to make proposals that are connected to our specific findings, in order to highlight the specific potential contribution of our study, regarding clinical practice.
Comment 23: conclusion - The conclusion summarizes the main findings appropriately but overstates the implications. Use more tentative language and avoid implying causality. The statement that THA leads to better QoL regardless of age contradicts earlier findings in the manuscript. Emphasize that psychological and social factors moderate recovery outcomes.
Response 23: We rewrote the conclusion in which we kept the summarizing of the main findings and we add some more specific proposals regarding future research, based on our findings (Lines 336-347). We deleted the statement that you report, and we avoided expressions that imply causality.
Response to Comments on the Quality of English Language
Point 1: The English could be improved to more clearly express the research.
Response 1: We conducted a careful language editing.

Reviewer 2 Report
Comments and Suggestions for Authors
Dear authors,
The topic you address in this study, namely the influence of kinesiophobia, social support, and socio-demographic characteristics on the quality of life in patients who have undergone total hip arthroplasty (THA) due to osteoarthritis, is timely and insufficiently explored in the existing literature. Moreover, this study brings a significant contribution, especially by integrating the psychosocial dimension into the evaluation of postoperative outcomes, and represents a useful contribution in the field of orthopedics and rehabilitation, suggesting the need for broader interventions that include social support and psychological assistance during postoperative recovery.
However, I believe the paper has some shortcomings, which I would like to mention as follows:
- In the abstract, in the results section, lines 26–33, I believe relevant statistical data should be presented.
- In the introduction section, there are several statements that are not supported by references. Thus, in lines 40–42, 44–46, 46–48, 49–50, 56–57, 60–61, 64–67, 67–69, 77, 77–79, please address this issue!
- Also in the introduction, I recommend you highlight more clearly why it is important for the two factors — kinesiophobia and social support — to be investigated together, not just individually, in the context of post-THA recovery. Additionally, the gaps in the existing literature should be more clearly stated.
- In subsection 2.1. Settings and Procedure, I believe the inclusion and exclusion criteria for the study need to be defined much more clearly. Also, the statement “…with sufficient knowledge of Greek language” is unclear to me. The hospital is in Athens, Greece, so the majority of patients are presumably Greek nationals. Therefore, I do not understand this statement. Please explain it!
- Also in subsection 2.1. Settings and Procedure, the period during which this study was conducted is not specified. Furthermore, the sample selection method is not mentioned: was it consecutive, random, or convenience sampling? Please add this information.
- Also in the methods section, I believe it would be useful to mention whether a sample size calculation was performed to justify the statistical power of the study. Additionally, I think the classification thresholds used to distinguish between high and low levels, both for TSK and OSSS-3, should be clarified.
- Also in the methods section, it should be clearly specified whether variables such as comorbidities or preoperative functional status were taken into account. Also, details about the postoperative period should be provided. At what point were the patients evaluated? At 3 months? At 6 months? Please clarify these aspects!
- In subsection 2.3. Ethical Issues, there is no mention of how data were collected from patients with mental health problems. How was consent obtained in such cases? Who completed the data, the patient or their representative? Perhaps these uncertainties would not have arisen if the inclusion and exclusion criteria had been better defined.
- Lines 142–146 – I believe this information should be synthesized, and much of it moved to the methods section.
- Regarding the reporting of results, I recommend adding confidence intervals (CI) to better understand the precision of the estimates. Additionally, the explanatory power of each model, namely R², should be reported.
- In the discussion section, you should compare the data obtained in your study with those obtained in other studies. This comparative analysis highlights the original contribution of your study compared to others. Also, some paragraphs in this section are repetitive. Please eliminate these repetitions. Moreover, the practical implications for doctors, psychologists, etc., should be mentioned, as in my opinion they are not sufficiently developed.
- Under limitations, it might be worth mentioning that the preoperative condition of the patients was not evaluated, and that the participants came from a single hospital and a single urban center. I believe these factors limit the accurate estimation of the intervention's impact.
Author Response
Response to Reviewer 2 Comments
Summary
Thank you very much for taking the time to review this manuscript. Please find the detailed responses below and the corresponding revisions highlighted in the re-submitted files.
Point-by-point response to Comments and Suggestions for Authors
Comments 1: The topic you address in this study, namely the influence of kinesiophobia, social support, and socio-demographic characteristics on the quality of life in patients who have undergone total hip arthroplasty (THA) due to osteoarthritis, is timely and insufficiently explored in the existing literature. Moreover, this study brings a significant contribution, especially by integrating the psychosocial dimension into the evaluation of postoperative outcomes, and represents a useful contribution in the field of orthopedics and rehabilitation, suggesting the need for broader interventions that include social support and psychological assistance during postoperative recovery.
Response 1: We would like to thank you for your suggestions that allowed us to make several improvements. Our responses to your comments are presented point by point below. All changes in the manuscript are highlighted in red.
Comment 2: In the abstract, in the results section, lines 26–33, I believe relevant statistical data should be presented.
Response 2: Thank you for your comment. We rewrote the results in the abstract and we focused only on the statistically significant data regarding the relationships between the independent variables and T-QoL subscales (lines 27-34). However, we did not add statistical values in the results due to words limitation (up to 250 words).
Comment 3: In the introduction section, there are several statements that are not supported by references. Thus, in lines 40–42, 44–46, 46–48, 49–50, 56–57, 60–61, 64–67, 67–69, 77, 77–79, please address this issue!
Response 3: Thank you for your suggestion. The main reason for the lack of references is in this part has to do with the fact that the references regarding the main variables/aspects of our study are in many cases the same and as such the referred in most cases at the end. However, in order to support our statements and arguments more clearly, we followed your suggestion. We also add three more references to support our statements.
Comment 4: Also in the introduction, I recommend you highlight more clearly why it is important for the two factors — kinesiophobia and social support — to be investigated together, not just individually, in the context of post-THA recovery. Additionally, the gaps in the existing literature should be more clearly stated.
Response 4: Thank you for your comments. We expanded the last paragraph of the introduction, where we present the aims of the study (Lines 102-122). There, we tried to explain better our contribution and to provide more specific arguments for the added value of our study, by testing two different “processes” (i.e., a negative which begins from kinesiophobia, and a negative which begins from social support) in the same population. Additionally, we also tested the interaction between social support and kinesiophobia, to test the buffering effects of social support in the negative relationship between kinesiophobia and aspects of quality of life. Finaly, we add some information in which we summarized the findings of the relevant literature regarding the role of sociodemographic, which, although do not consist the main focus of our study, they are also be tested, as far as their relationship with quality of life after THA (Lines 91-101).
Comment 5: In subsection 2.1. Settings and Procedure, I believe the inclusion and exclusion criteria for the study need to be defined much more clearly. Also, the statement “…with sufficient knowledge of Greek language” is unclear to me. The hospital is in Athens, Greece, so the majority of patients are presumably Greek nationals. Therefore, I do not understand this statement. Please explain it!
Response 5: Thank you for pointing this. The criterion of the language was set for the case of immigrants that live in Greece many years and have access to national system. However, the statement may confuse the reader, and thus, was
rephrased to: “Inclusion criteria were as follows: adult patients subjected to elective THA due to hip OA, with sufficient knowledge of Greek language (i.e., in a possible case of immigrant participants)” (Lines 132-136).
Comment 6: Also in subsection 2.1. Settings and Procedure, the period during which this study was conducted is not specified. Furthermore, the sample selection method is not mentioned: was it consecutive, random, or convenience sampling? Please add this information.
Response 6: The period during which the study was conducted was added in the text: ““Data were collected between June of 2021 and February of 2022, while at the time of the survey each participant has completed a five-month of a post-operative period…” (Lines 130-132). Moreover, we added the information regarding the sampling method: “The study population consisted of a convenience sample” (Line 128).
Comment 7: Also in the methods section, I believe it would be useful to mention whether a sample size calculation was performed to justify the statistical power of the study. Additionally, I think the classification thresholds used to distinguish between high and low levels, both for TSK and OSSS-3, should be clarified.
Response 7: It was calculated that with the sample of 125 participants the study will be have 95% power to conduct a linear regression model at a significance level of 0.05 and for effect sizes equal to 0.20 or greater.
Comment 8: Also in the methods section, it should be clearly specified whether variables such as comorbidities or preoperative functional status were taken into account. Also, details about the postoperative period should be provided. At what point were the patients evaluated? At 3 months? At 6 months? Please clarify these aspects!
Response 8: Thank you for your suggestions. Indeed, this is crucial to be reported. We clarified what you mention in the Settings and Procedure sub-section “Data were collected between June of 2021 and February of 2022, while at the time of the survey each participant has completed a five-month of a post-operative period, so as the surgery could be perceived eligible” (Lines 130-132).
Comment 9: In subsection 2.3. Ethical Issues, there is no mention of how data were collected from patients with mental health problems. How was consent obtained in such cases? Who completed the data, the patient or their representative? Perhaps these uncertainties would not have arisen if the inclusion and exclusion criteria had been better defined.
Response 9: Thank you for pointing this. In the study did not participate patients with a mental-health diagnosis or diagnosed cognitive impairment. These were clarified in the re-written text regarding inclusion and exclusion criteria: “Inclusion criteria were as follows: adult patients subjected to elective THA due to hip OA, with sufficient knowledge of Greek language (i.e., in a possible case of immigrant participants). Exclusion criteria were: the absence of a diagnosis of mental and/or cognitive impairment problems” (Lines 132-136).
Comment 10: Lines 142–146 – I believe this information should be synthesized, and much of it moved to the methods section.
Response 10: Because of the Table 1 presentation in the results, we kept this information in this sub-section. However, we moved the number of the participants in the methods, while we also added the response rate (see Settings and Procedure subsection - Lines 135-136). We also partly tried to synthesized the information by shortening the text since the information is provided in the Table 1.
Comment 11: Regarding the reporting of results, I recommend adding confidence intervals (CI) to better understand the precision of the estimates. Additionally, the explanatory power of each model, namely R², should be reported.
Response 11: R-squared values and 95% Cis were added in the regression results (see Tables 2-5).
Comment 12: In the discussion section, you should compare the data obtained in your study with those obtained in other studies. This comparative analysis highlights the original contribution of your study compared to others.
Response 12: Thank you for the comment. We reformed the discussion by following the order bellow: First, we summarized the findings (1st paragraph), then we discussed our findings, beginning from the relationship between quality-of-life aspects and kinesiophobia, social support sociodemographic, and mental health (2nd to 5th paragraph), next the practical implications, based on our findings (6th paragraph) and finally the limitations (7th paragraph). In the discussion we tried to highlight the mechanisms that explains the significant relationships and provide more specific arguments regarding the interpretation of each finding.
Comment 13: Also, some paragraphs in this section are repetitive. Please eliminate these repetitions.
Response 13: See the comment above. By reconstructing the discussion, we tried to eliminate previous repetitions.
Comment 14: Moreover, the practical implications for doctors, psychologists, etc., should be mentioned, as in my opinion they are not sufficiently developed.
Response 14: Thank you for your suggestions. At the end of the discussion, and before limitations, we added a new paragraph with practical implications (Lines 303-324). There, by considering, also, our specific findings, we tried to provide some more details regarding potential interventions which may enhance quality of life after THA, through kinesiophobia management and social support enhancement. In the text we, also, tried to discriminate the role of different health professionals.
Comment 15: Under limitations, it might be worth mentioning that the preoperative condition of the patients was not evaluated, and that the participants came from a single hospital and a single urban center. I believe these factors limit the accurate estimation of the intervention's impact.
Response 15: Thank you for your suggestion. We added the specific information in the limitations (Lines 325-334).

Reviewer 3 Report
Comments and Suggestions for Authors
The manuscript presents a relevant and timely investigation into the psychological and social factors influencing quality of life after total hip arthroplasty (THA). The topic is of significant clinical importance and is addressed with methodological rigor. Below are a few specific suggestions to further enhance the clarity and impact of your work:
1. Introduction:
-
The introduction provides a solid background and clearly states the study's purpose.
-
Suggestion: You may consider briefly elaborating on the potential mechanisms through which kinesiophobia and social support affect recovery outcomes in THA patients. This would help to frame your research questions within a more comprehensive biopsychosocial model.
2. Methods:
-
The methodology is sound and replicable. The use of validated instruments strengthens the study’s reliability.
-
Suggestion: Clarify whether patients were all at a similar time point post-surgery when completing the survey, as time since surgery could influence reported outcomes. If variability existed, consider noting this as a limitation.
3. Results:
-
The results are presented with well-structured tables and appropriate statistical analysis.
-
Suggestion: Where feasible, highlight clinically meaningful effect sizes or thresholds, especially to kinesiophobia scores, to aid interpretation for applied health professionals.
4. Discussion and Conclusion:
-
The discussion appropriately interprets findings and relates them to existing literature.
-
Suggestion: Expand briefly on how the findings might inform future clinical interventions, such as targeted rehabilitation programs or psychosocial support post-THA. This would strengthen the translational value of your study.
5. Language and structure:
-
The manuscript is written with a good academic tone and fluent English. No major revisions are needed.
This is a well-executed study that adds meaningful insights into the role of psychosocial variables in post-surgical recovery. Hope the suggestions provided support the further enhancement of your manuscript.
Author Response
Response to Reviewer 3 Comments
Summary
Thank you very much for taking the time to review this manuscript. Please find the detailed responses below and the corresponding revisions highlighted in the re-submitted files.
Point-by-point response to Comments and Suggestions for Authors
Comments 1: The manuscript presents a relevant and timely investigation into the psychological and social factors influencing quality of life after total hip arthroplasty (THA). The topic is of significant clinical importance and is addressed with methodological rigor. Below are a few specific suggestions to further enhance the clarity and impact of your work:
Response 1: We would like to thank you for your comments and suggestions. Our responses to your suggestions are presented point by point, below. All changes in the manuscript are highlighted in red.
Comment 2: 1. Introduction: The introduction provides a solid background and clearly states the study's purpose. Suggestion: You may consider briefly elaborating on the potential mechanisms through which kinesiophobia and social support affect recovery outcomes in THA patients. This would help to frame your research questions within a more comprehensive biopsychosocial model.
Response 2: Thank you for your suggestion. We changed the last paragraph of the introduction (Lines 102-122). There, we tried to explain better the added value of our study, by testing two different “processes” (i.e., a negative which begins from kinesiophobia, and a negative which begins from social support) in the same population. Moreover, we also tested the interaction between social support and kinesiophobia, to test the buffering effects of social support in the negative relationship between kinesiophobia and aspects of quality of life. With this way we tried to highlight the mechanisms through which kinesiophobia and social support affect recovery outcomes in THA patients, and present a more comprehensive narrative.
Comment 3: 2. Methods: The methodology is sound and replicable. The use of validated instruments strengthens the study’s reliability. Suggestion: Clarify whether patients were all at a similar time point post-surgery when completing the survey, as time since surgery could influence reported outcomes. If variability existed, consider noting this as a limitation.
Response 3: Thank you for pointing this out. Data were collected in an 8-month period, while the participants were invited after a five-month post-operative period. In the 2.1. Settings and Procedure sub-section we added the information: “Data were collected between June of 2021 and February of 2022, while at the time of the survey each participant has completed a five-month of a post-operative period, so as the surgery could be perceived eligible” (Lines 130-132).
Comment 4: 3. Results: he results are presented with well-structured tables and appropriate statistical analysis. Suggestion: Where feasible, highlight clinically meaningful effect sizes or thresholds, especially to kinesiophobia scores, to aid interpretation for applied health professionals.
Response 4: Thank you for your comment. We run again the analyses. Kinesiophobia and social support were analyzed as continuous (and not dichotomous) variables. The new results, in which we do not present thresholds, are presented in the text.
Comment 5: 4. Discussion and Conclusion: The discussion appropriately interprets findings and relates them to existing literature. Suggestion: Expand briefly on how the findings might inform future clinical interventions, such as targeted rehabilitation programs or psychosocial support post-THA. This would strengthen the translational value of your study.
Response 5: Thank you for the useful proposal. At the end of the discussion, and before limitations, we added a new paragraph with practical implications (Lines 303-324). There, by taking into account and our specific findings, we tried to provide some more details regarding potential interventions which may enhance quality of life after THA, through kinesiophobia management and social support enhancement.
Comment 6: 5. Language and structure: The manuscript is written with a good academic tone and fluent English. No major revisions are needed. This is a well-executed study that adds meaningful insights into the role of psychosocial variables in post-surgical recovery. Hope the suggestions provided support the further enhancement of your manuscript.
Response 6: Thank you very much for your useful proposals and for the positive feedback.

Round 2
Reviewer 1 Report
Comments and Suggestions for Authors
Dear Authors,
Thank you very much for allowing me to review the manuscript again. You have satisfactorily addressed the issues raised.
Reviewer 2 Report
Comments and Suggestions for Authors
Dear authors,
Thank you so much for considering my suggestions. I really think your research paper has improved a lot.
Best of luck with your future research projects!